# Simulation of Binodal and Spinodal Curves of Phase State Diagrams for Binary Polymer Systems

**DOI:** 10.3390/polym14132524

**Published:** 2022-06-21

**Authors:** Anatoly E. Chalykh, Uliana V. Nikulova, Vladimir K. Gerasimov

**Affiliations:** Laboratory of Structural and Morphological Investigations, Frumkin Institute of Physical Chemistry and Electrochemistry Russian Academy of Sciences (IPCE RAS), Leninsky pr. 31-4, 119071 Moscow, Russia; ulianan@rambler.ru (U.V.N.); vladger@mail.ru (V.K.G.)

**Keywords:** phase state diagram, mixing thermodynamics, simulation, polystyrene, polymethyl methacrylate, polybutadiene

## Abstract

A new approach is proposed for simulating binodal and spinodal curves of phase diagrams for binary polymer systems. It is shown that the Flory–Huggins theory makes it possible to predict phase behavior in a wide range of temperatures and concentrations based on limited data on the components’ solubility. The approbation data of the technique are presented in the example of PS–PB and PS–PMMA systems, for which generalized phase diagrams are constructed.

## 1. Introduction

The construction of phase state diagrams has been and remains a priority task both in the thermodynamics of polymer solutions and melts and in polymer materials science [1,2,3,4,5]. It is known that the boundary curves, liquidus, and solidus lines are of fundamental importance for the construction and interpretation of phase state diagrams since they determine not only the type of phase equilibrium but also the arias of formation of various phase structures in the temperature-concentration regions.

Thus, the binodal curve in systems with amorphous equilibrium separates the region of the true solutions from the region of the two-phase state. The region located between the binodal curve and the spinodal curve makes it possible to identify the position on the diagram of the region of the metastable and labile solutions. Finally, in the middle region of the diagram, there are boundary curves framing the region of phase reversal. The construction of generalized diagrams [6] makes it possible to determine the regions of heterogeneous fluctuations, the position of critical points, and the regions of glass transition, viscous flow, and the thermal stability of components. Thus, with a generalized phase state diagram, it is possible to predict the structural and morphological transformations that occur under certain conditions of processing (high-temperature regimes with rapid cooling), the exploitation of materials, and also during the synthesis of copolymers. It is obvious that to solve some material science problems, it is necessary to expand the temperature and concentration intervals for plotting state diagrams.

The importance of predicting and constructing state diagrams of polymer–polymer systems is confirmed by the fact that two independent directions are currently being developed in this area of the physical chemistry of polymers. The first—theoretical simulation—concerns the statistical thermodynamics of solutions and polymer blends. These are the classical Flory–Huggins theory, the new Flory theory, the Sanchez–Lacombe theory, and Prigogine–Paterson theory [1,2,3,4,5], the statistical associating fluid theory (SAFT) [7], the polymer reference interaction site model (PRISM) [8], and the lattice cluster theory (LCT) [9]. Recently, the adapted mean field theory [10], the hybrid molecular dynamics-Monte Carlo method [11], the self-consistent field theory (SCFT) [12], and theoretically informed Langevin dynamics [13] have been used in diagram simulation. All of these theories make it possible to obtain analytical dependences of the free energy of mixing components on the composition of systems and to determine the position of binodal and spinodal boundary curves and critical states in the framework of classical thermodynamics. It is interesting to note that comparative studies of the prediction of the above statistical theories of binary polymer compatibility have shown that the classical Flory–Huggins theory has a good ability to predict phase equilibrium in polymer solutions and melts [2].

The second direction is associated with the independent determination of the Flory–Huggins parameter based on the data of inverse gas chromatography, polymer solutions, intrinsic viscosity, and sorption [3,14,15,16,17]. That is, the prediction of the boundary curves here is partly based on preliminary experimental data for determining the interaction parameter. Extrapolation of the obtained physicochemical characteristics to the temperature ranges of interest to researchers also implies the calculation of the free energy of mixing and the subsequent expansion of the investigated range of the components’ compatibility. It is obvious that the construction of such diagrams is identical to the prediction of segments of phase-state diagrams. In this case, the biggest difficulties arise with the construction of binodal curves due to the uncertainty in estimating the compositions of coexisting phases using binodal equations.

In this work, we chose the second approach, but we estimated the interaction parameters not by third-party methods, but from the data on the solubility of the components in a limited range. At the same time, we propose a more efficient and convenient method for numerically determining the compositions of coexisting phases and constructing the position of binodal and spinodal curves in a wide range of temperatures and molecular weights of binary system components. This approach makes it possible to construct generalized phase diagrams for different types of polymer–polymer systems, which has been tested on a number of examples.

## 2. Theoretical Methodology

The Flory–Huggins theory of polymer solutions [1,2,3,4,5] is based on the simplest lattice model of an athermal solution, which contains assumptions about the volumes of macromolecular chain segments and their relative mobility, which is associated with the combinatorial entropy of all possible permutations within such a lattice.

Traditionally, in the framework of the Flory–Huggins theory, the free energy of mixing two polymers is represented as follows:(1)$$\Delta G=RT\frac{V}{V_{r}}\left[\frac{\varphi_{1}}{r_{1}}ln\varphi_{1}+\frac{\varphi_{2}}{r_{2}}ln\varphi_{2}+\chi_{12}\varphi_{1}\varphi_{2}\right]$$
where R is the universal gas constant, T is the temperature, V is the polymer mixing volume, $V_{r}$ is the reference volume, usually taken as the molar volume of a repeating unit in the system (in calculations, a reference volume of 100 cm^3^/mol is traditionally used [2]), $\varphi_{1}$ and $\varphi_{2}$ are the volume fractions of the first and second components in the system, $r_{1}$ and $r_{2}$ are their polymerization degrees, $\chi_{12}$ is the Flory–Huggins interaction parameter.

Expressions for chemical potentials are as follows:(2)$$\Delta \mu_{1}=RT\left[\frac{ln\varphi_{1}}{r_{1}}+\left(\frac{1}{r_{1}}-\frac{1}{r_{2}}\right)\varphi_{2}+\chi_{12}\varphi_{2}^{2}\right]\;\Delta \mu_{2}=RT\left[\frac{ln\varphi_{2}}{r_{2}}+\left(\frac{1}{r_{2}}-\frac{1}{r_{1}}\right)\varphi_{1}+\chi_{21}\varphi_{1}^{2}\right]$$

The conditions at the critical point are as follows:(3)$$\varphi_{1,cr}=\frac{\sqrt{r_{2}}}{\sqrt{r_{1}}+\sqrt{r_{2}}}\;\varphi_{2,cr}=\frac{\sqrt{r_{1}}}{\sqrt{r_{1}}+\sqrt{r_{2}}}$$
(4)$$\chi_{cr}=\frac{1}{2}{\left(\frac{1}{\sqrt{r_{1}}}+\frac{1}{\sqrt{r_{2}}}\right)}^{2}$$

Spinodal concentrations are defined as the points at which the second derivative of the free energy of mixing with respect to concentration is zero. The spinodal equation takes the form:(5)$$\frac{1}{r_{1}\varphi_{1,s}}+\frac{1}{r_{2}\varphi_{2,s}}-2\chi =0$$
where χ is the averaged value of the pair interaction parameter.

The binodal equations are calculated from the condition that the chemical potentials of both polymers are equal in two coexisting phases:(6)$$\frac{ln\left(\varphi_{1}^{\prime }\right)}{r_{1}}+\left(\frac{1}{r_{1}}-\frac{1}{r_{2}}\right)\varphi_{2}^{\prime }+\chi_{12}{\left(\varphi_{2}^{\prime }\right)}^{2}=\frac{ln\left(\varphi_{1}^{\prime \prime }\right)}{r_{1}}+\left(\frac{1}{r_{1}}-\frac{1}{r_{2}}\right)\varphi_{2}^{\prime \prime }+\chi_{12}{\left(\varphi_{2}^{\prime \prime }\right)}^{2}\;\frac{ln\left(\varphi_{2}^{\prime }\right)}{r_{2}}+\left(\frac{1}{r_{2}}-\frac{1}{r_{1}}\right)\varphi_{1}^{\prime }+\chi_{21}{\left(\varphi_{1}^{\prime }\right)}^{2}=\frac{ln\left(\varphi_{2}^{\prime \prime }\right)}{r_{2}}+\left(\frac{1}{r_{2}}-\frac{1}{r_{1}}\right)\varphi_{1}^{\prime \prime }+\chi_{21}{\left(\varphi_{1}^{\prime \prime }\right)}^{2}$$
where ′ and ″ refer to different coexisting phases.

For thermodynamic analysis of the obtained experimental data on binodal curves (or their fragments), the presented equations lead to the following expressions for the pair interaction parameter:(7)$$\chi_{12}=\frac{ln\left(\varphi_{1}^{\prime \prime }\right)-ln\left(\varphi_{1}^{\prime }\right)}{r_{1}\left({\left(\varphi_{2}^{\prime }\right)}^{2}-{\left(\varphi_{2}^{\prime \prime }\right)}^{2}\right)}-\left(\frac{1}{r_{1}}-\frac{1}{r_{2}}\right)\frac{1}{\varphi_{2}^{\prime }+\varphi_{2}^{\prime \prime }}$$
(8)$$\chi_{21}=\frac{ln\left(\varphi_{2}^{\prime \prime }\right)-ln\left(\varphi_{2}^{\prime }\right)}{r_{2}\left({\left(\varphi_{1}^{\prime }\right)}^{2}-{\left(\varphi_{1}^{\prime \prime }\right)}^{2}\right)}-\left(\frac{1}{r_{2}}-\frac{1}{r_{1}}\right)\frac{1}{\varphi_{1}^{\prime }+\varphi_{1}^{\prime \prime }}$$

The averaged value of the pair interaction parameter, assuming the absence of its concentration dependence, can be represented as follows:(9)$$\chi =\frac{\frac{ln\left(\varphi_{1}^{\prime \prime }/\varphi_{1}^{\prime }\right)}{r_{1}}-\frac{ln\left(\varphi_{2}^{\prime \prime }/\varphi_{2}^{\prime }\right)}{r_{2}}}{2\left(\varphi_{2}^{\prime }-\varphi_{2}^{\prime \prime }\right)}$$

Figure 1 shows a scheme of the methodology for the thermodynamic analysis of binary polymer systems and the construction of generalized phase diagrams. At the initial stage, fragments of binodal curves are plotted based on the experimentally obtained data on the solubility of polymers in each other (Figure 1a). Then, according to the compositions of the coexisting phases $\varphi_{1}^{\prime }$ and $\varphi_{1}^{\prime \prime }$ at different temperatures *T*, using Equation (9), the numerical values of the pair interaction parameter χ are calculated, and its temperature dependence is plotted (Figure 1b). Extrapolation of this dependence to $\chi_{cr}$ makes it possible to obtain information about the critical solution temperature of the components (in this case, the upper critical solution temperature — UCST). In coordinates χ=A+B·1T it is possible to identify χ data in a wide range of temperatures at *T* ≥ *T*_ex_, *T* ≤ *T*_ex_, *T* ≥ USCT. Based on these data, using Equations (5) and (6), the boundary spinodal (dashed line) and binodal (solid line) curves are calculated, and a generalized phase diagram is constructed (Figure 1c) in the selected temperature range, which traditionally denotes the region of homogeneous states (I), the region of labile structures (II), and the region of metastable states (III).

We should consider in more detail the procedure for determining data on the coexisting phase compositions. It is known that the concentration dependence of free energy for a polymer–polymer system at a temperature *T* ≤ UCST, corresponding to the limited mixing of components, is a complex curve with two minima and two inflection points (Figure 2a). By drawing a common tangent to the dependence ΔG−φ, corresponding to the equality of chemical potentials, it can determine the concentrations at the points of the curves contact (points 2) about thus assess the information about the coexisting phases compositions for the binodal curve ($\varphi_{b}^{\prime }$ and $\varphi_{b}^{\prime \prime }$), and in inflection points (points 3) receive information for constructing a spinodal curve ($\varphi_{c}^{\prime }$ and $\varphi_{c}^{\prime \prime }$). Numerical determination of binodal and spinodal concentrations of more or less symmetric concentration dependences of the free mixing energy does not cause any difficulties [18]. This situation is realized in cases where the molecular weights of the components do not differ much. When analyzing a similar dependence for a binary system with significantly different molecular weights of components—for example, a polymer solution, a polymer–plasticizer mixture, a polymer–oligomer system, and even a high-molecular polymer–low-molecular polymer blend—significant difficulties arise in the standard approach. It can be seen (Figure 2b) that in this case the dependence $\Delta G_{m}\left(\varphi \right)$ becomes sharply asymmetric, one of the binodal concentrations shifts to the region of small values, the curve flattens out in this place, which greatly complicates the numerical determination of the phase composition. In this case, the direct mathematical solution of the system of Equation (6) leads to a multiplicity of solutions that are difficult to automatically limit.

Analyzing the principle of equality of chemical potentials in different phases and its interpretation as a common tangent to the dependence $\Delta G_{m}\left(\varphi \right)$, we can assume that there are the same values at two points of the concentration dependences of chemical potentials. This means that the dependence $\Delta \mu_{2}\left(\Delta \mu_{1}\right)$ must have a self-intersection point (two points of the dependence must have the same value). For a single-phase system, this dependence has no self-intersection points, and for a two-phase system, it has a single point corresponding to the equality of chemical potentials in different phases, i.e., thermodynamic condition of phase equilibrium. Figure 3 shows such a schematic concentration dependence of chemical potentials where the values of $\Delta \mu_{1}$ and $\Delta \mu_{2}$ correspond to the same value of the concentration of the components. Therefore, the enumeration of concentrations in the range from 0 to 1 gives a set of pairs of chemical potentials for this dependence. The point of self-intersection (1) determines, in this case, the concentrations of the binodal curve.

Accordingly, when constructing a generalized phase diagram, the spinodal Equation (5) can be used to determine the composition of the coexisting phases of the spinodal curve. The transformation of this equation allows us to express its roots in the following form:(10)$$\varphi_{1,s}=\frac{-\left(r_{1}-r_{2}-2\chi r_{1}r_{2}\right)\pm \sqrt{{\left(r_{1}-r_{2}-2\chi r_{1}r_{2}\right)}^{2}-4\cdot 2\chi r_{1}r_{2}\cdot r_{2}}}{2\cdot 2\chi r_{1}r_{2}}$$

To find the binodal points, we propose using the following mathematical algorithm:

-Using the iteration method, it can be calculated as a set of $\mu_{1}$ and $\mu_{2}$ values for each temperature by mean of enumeration of successive concentrations $\varphi_{1}$ in the range of 0 to 1, with a step of 0.01;-build the dependence of $\mu_{2}$ on $\mu_{1}$, where each point corresponds to a certain set of concentrations (Figure 3);-determine the concentration range in which the self-intersection point of such dependence is localized; then, decreasing the iteration step in concentration by one-tenth, we successively narrow the interval of the self-intersection point ten times to establish the self-intersection point with an accuracy of $\varphi_{1}$ of the order of 0.001 and accept the concentrations for chemical potentials at this point as $\varphi_{1}^{\prime }$ and $\varphi_{1}^{\prime \prime }$, corresponding to the compositions of coexisting phases binodal curve;-Determine the position of the critical point on the phase diagram based on the calculated value of the concentration at the critical point according to Equation (3) and the critical temperature according to the point of intersection of the temperature dependence χ and the calculated value $\chi_{cr}$ according to Equation (4) (as shown in Figure 1b).

## 3. Experiment

The following monodisperse polymers were used as objects for approbation of the methodology: polystyrene (PS) with an average molecular weight of *M*_w_ = 37.8 kDa and *M*_w_ = 0.8 kDa (PS-1 and PS-2, respectively) manufactured by Waters Associates, polybutadiene (PB) with *M*_w_ = 61.6 kDa (Aldrich), and polymethyl methacrylate (PMMA) with *M*_w_ = 89 kDa (Glass).

The solubility data of the components were obtained using the optical interferometry method [19]. Pressed polymer films about 5 × 10 mm in size were placed between two optically transparent glasses, on the inside of which a translucent metal layer (nichrome) was deposited by thermal vacuum deposition. Glasses with polymers between them were clamped in a temperature-controlled cell (±1 K) at a temperature slightly above the glass transition (melting) temperature of polymers to a variable thickness of 100–120 μm, fixed by special metal wedges. The polymers were in optical contact with the inner surfaces of the glasses. A digital camera through a microscope with standard magnifications fixed the interference from a light source (helium-neon laser with a wavelength of 632 nm), passing perpendicular to the film contact plane and forming a concentration profile of polymer interaction at the contact boundary. The measurements were carried out in the mode of a stepwise increase and decrease in temperature from 400 to 540 K. The temperature step was 20–40 K with holding at each step for at least 30–60 min, which makes it possible to obtain equilibrium data. The reproducibility of data in heating–cooling cycles allows us to speak about the equilibrium values of the compositions of coexisting phases at each temperature. This technique makes it possible to simultaneously observe the entire concentration range at the contact boundary of the two components, and the error in determining the compositions of the coexisting phases at each temperature is about 5%. The procedure for conducting the experiment and processing the interferograms did not differ from the traditional procedure [20,21,22,23].

## 4. Results and Discussion

To test the calculations using the proposed method, experimental results on solubility in the PS–PB and PS–PMMA systems were used.

Figure 4 shows fragments of binodal curves for the PB–PS-1 and PMMA–PS-2 systems. The molecular weights of the first pair do not differ much (*M*_w PB_/*M*_w PS-1_ = 1.6), but in the second case, they differ significantly (*M*_w PMMA_/*M*_w PS-2_ = 111.3). Thus, we tried to simulate two different cases for polymer–polymer systems according to the scheme in Figure 2. The data for both systems are characterized by limited solubility, in the range from 400 to 530 K for the PB–PS-1 system and from 460 to 520 K for the PMMA–PS-2 system. For both pairs, the experimental range of values was limited by the glass transition/melting temperatures from below and the thermal oxidative degradation from above. Nevertheless, it can be said that the solubility of the components increases with temperature growth, and it should expect the appearance of an upper critical solution temperature (UCST) in the higher temperatures’ region.

According to the method proposed above, using Equation (9), pair interaction parameters were calculated, and their temperature dependences were plotted (Figure 5). It can be seen that both systems demonstrate a linear change χ in the proposed coordinates, and interpolation of such a dependence is possible in the form χ=3.4748·(1/T)−0.0022 for the PB–PS-1 system and χ=142.78·(1/T)−0.1608 for the PMMA–PS-2 system. Such interpolation allows us to extrapolate data to a wide temperature range and obtain, firstly, information about the critical temperature from the intersection of the dependence χ−(1/T) and the calculated value $\chi_{cr}$ (as shown schematically in Figure 1b). In accordance with this, the UCST for the PB–PS-1 system is 628 K, and for the PMMA–PS-2 system, it is 598 K. The calculated values for the concentrations at the critical point $\varphi_{PS,\;cr}$ by Equation (3) can be given as 0.64 and 0.91 for each of the systems, respectively. Second, extrapolation of the temperature dependence of the interaction parameter enables us to use data on the χ−(1/T) ratio to determine the corresponding compositions of coexisting phases with any temperature step. The phase equilibrium data for the PMMA-PS system and the temperature dependence of the interaction parameter are in good agreement with the data obtained by the turbidity point method for this system [24,25], taking into account the difference in the PMMA molecular weight.

Figure 6 shows the calculated data on the free energy of mixing and chemical potentials in the studied systems. It can be seen that the PB–PS-1 system is characterized by a more symmetrical concentration dependence for ΔG (Figure 6a). The extrema of the function is explicitly defined both for the case $\varphi_{PS}\to 0$ and for the case $\varphi_{PS}\to 1$. For the PMMA–PS-2 system (Figure 6c) for $\varphi_{PS}\;\sim \;0.2$, the first extremum of such a dependence is uniquely determined, and the localization of the second extremum at $\varphi_{PS}\to 1$ cannot be determined. When passing to the concentration dependences of chemical potentials (Figure 6b,d), according to the proposed method, in both cases, we can identify the self-intersection point and determine the corresponding compositions of the coexisting phases for the binodal curves.

For both systems, the concentration dependences of the iteration method were used to obtain data on the coexisting phase compositions for the binodal curves, and the coexisting phase compositions of the spinodal curves were calculated using Equation (10). This made it possible to construct generalized phase diagrams for both the PB–PS-1 and PMMA–PS-2 systems (Figure 7a and Figure 7b, respectively). In both cases, the diagrams are characterized by UCST (gray dots), the position of which is localized in the region of high temperatures above the destruction temperatures of the components. However, compatibility information in this area, as well as in the area below the glass transition, is extremely important since blends in manufacturing processes can be obtained over a wide temperature range. In accordance with the difference in *M*_w_, the diagram for the PB–PS-1 system is more symmetrical than for the PMMA–PS-2 system. The calculated binodal (solid lines 1) and spinodal (dashed lines 2) curves correlate well with the initial experimental data on solubility (black dots) and accurately determine the position of homogeneous (I) and heterogeneous (II) regions, as well as the regions of metastable states (III) on the temperature-concentration field of the diagram. The dot-dashed line (3) shows the change in Alekseev’s diameter (rectilinear diameter rule) when approaching the UCST. In both cases, this dependence is curvilinear, and for the PMMA–PS-1 system, this is more pronounced. At the same time, for the PMMA–PS-2 system, the right part of the diagram coincides to a lesser extent with the calculated values according to the binodal curve but coincides with the spinodal one. This can be explained by the error in the experimental determination of the solubility of the components in this region of the interferograms.

It should also be noted that such an approach makes it possible to obtain generalized phase diagrams not only for systems with UCST but also for systems with a lower critical solution temperature (LCST). The technique was tested by us earlier on the classical system polystyrene–poly(vinyl methyl ether) (PVME) [22], which is characterized by the presence of LCST in the region of 370–470 K, depending on the PS molecular weight. In addition, the universality and simplicity of the mathematical apparatus of the Flory–Huggins theory and the presented approach make it possible to solve complex thermodynamic problems of simulation phase equilibrium diagrams in the presence of two critical solution temperatures. For example, for the PVME–water system [19], a generalized diagram was constructed with two LCST in the region of 290–310 K, and it was shown that part of this diagram represents the equilibrium between PVME and its complex with water, and the other part of the diagram is the equilibrium between PVME and water after the decomposition of the complex at an increase in temperature.

## 5. Conclusions

A method for calculating the coexisting phase compositions of binodal curves through the concentration dependences of chemical potentials in the framework of the Flory–Huggins theory was also proposed, using the example of the PS–PB and PS–PMMA systems. This made it possible to approach the construction of generalized phase-state diagrams with higher accuracy in cases where the critical point lies in the region of concentrated solutions due to a large difference in the molecular weights of the components. According to the proposed approach, there is no difficulty in determining the minima on the concentration dependence of the free energy of mixing since it is possible to fix the concentrations when the chemical potentials are equal.

The versatility and simplicity of the approach makes It possible to construct generalized phase diagrams containing binodal and spinodal curves for polymer–solvent, polymer–oligomer, polymer–polymer, and polymer–copolymer interactions characterized by both UCST and LCST.

## Figures and Tables

**Figure 1 polymers-14-02524-f001:**
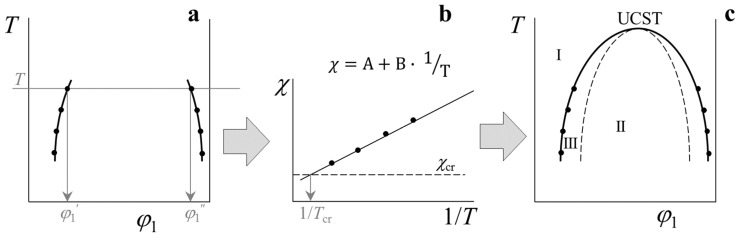
Method for simulation phase diagrams from fragments of binodal curves in the framework of the Flory–Huggins theory: (**a**)—initial data on solubility, (**b**)—temperature dependence of χ, (**c**)—simulation of a generalized phase state diagram. Black dots—experimental data, solid thick lines—binodal curves, dashed line—spinodal curve. (See text for details).

**Figure 2 polymers-14-02524-f002:**
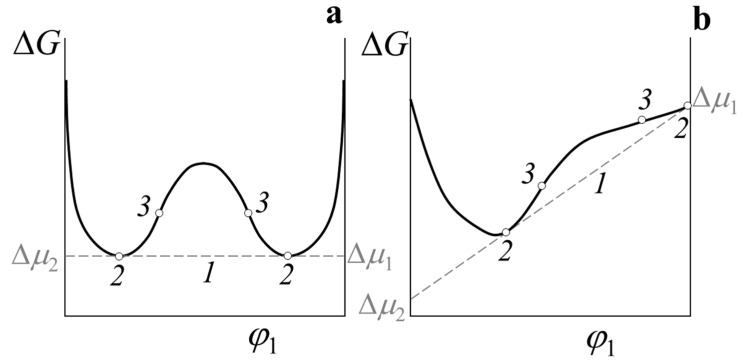
Concentration dependence of the free energy of mixing for symmetric (**a**) and asymmetric (**b**) cases. Common tangent (1), binodal (2), and spinodal (3) concentrations.

**Figure 3 polymers-14-02524-f003:**
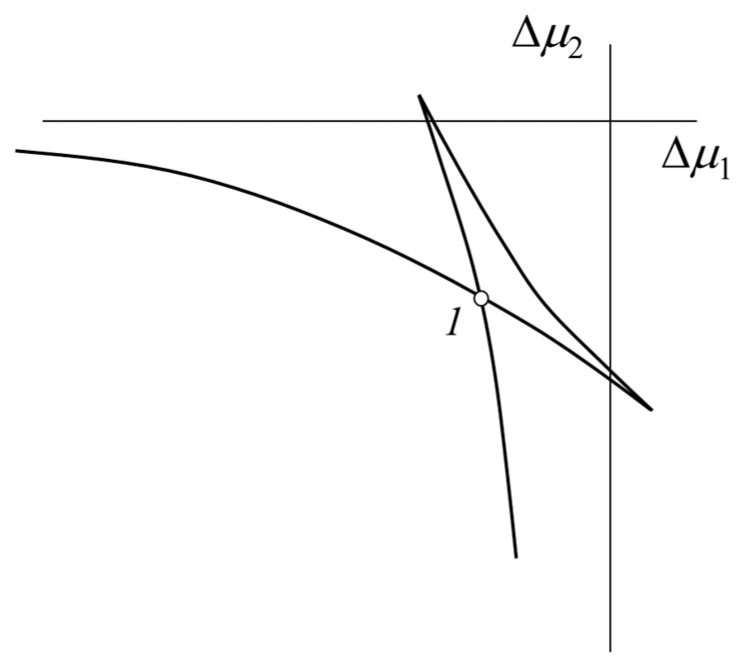
Graphical representation for the self-intersection point (1) of the concentration dependences of the chemical potentials.

**Figure 4 polymers-14-02524-f004:**
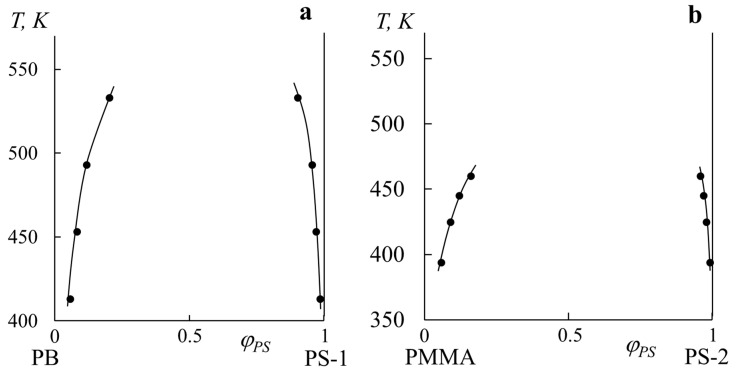
Fragments of binodal curves for PB–PS-1 (**a**) and PMMA–PS-2 (**b**) systems obtained by optical interferometry.

**Figure 5 polymers-14-02524-f005:**
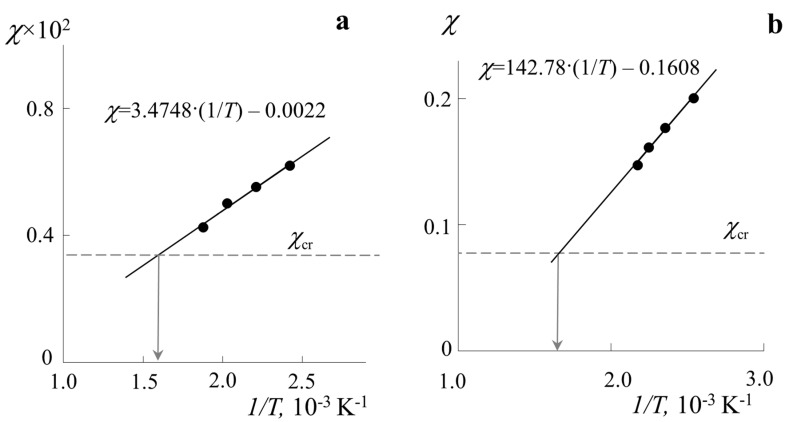
Temperature dependences of the Flory–Huggins parameter calculated by Equation (9) for PB–PS-1 (**a**) and PMMA–PS-2 (**b**) systems.

**Figure 6 polymers-14-02524-f006:**
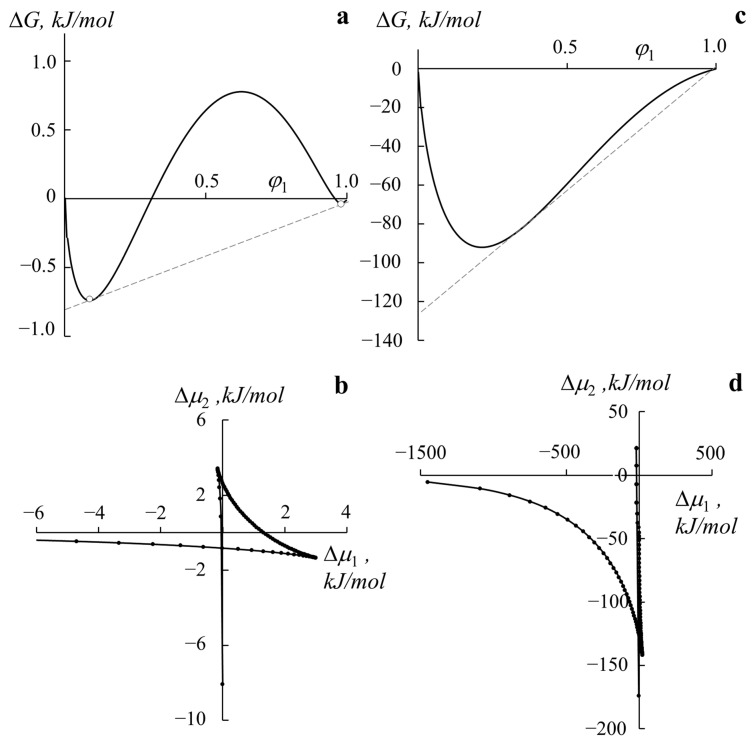
Concentration dependences of Gibbs free energy (**a**,**c**) and chemical potentials (**b**,**d**) for the PB–PS-1 system (**a**,**b**) at 440 K and the PMMA–PS-2 system (**c**,**d**) at 500 K.

**Figure 7 polymers-14-02524-f007:**
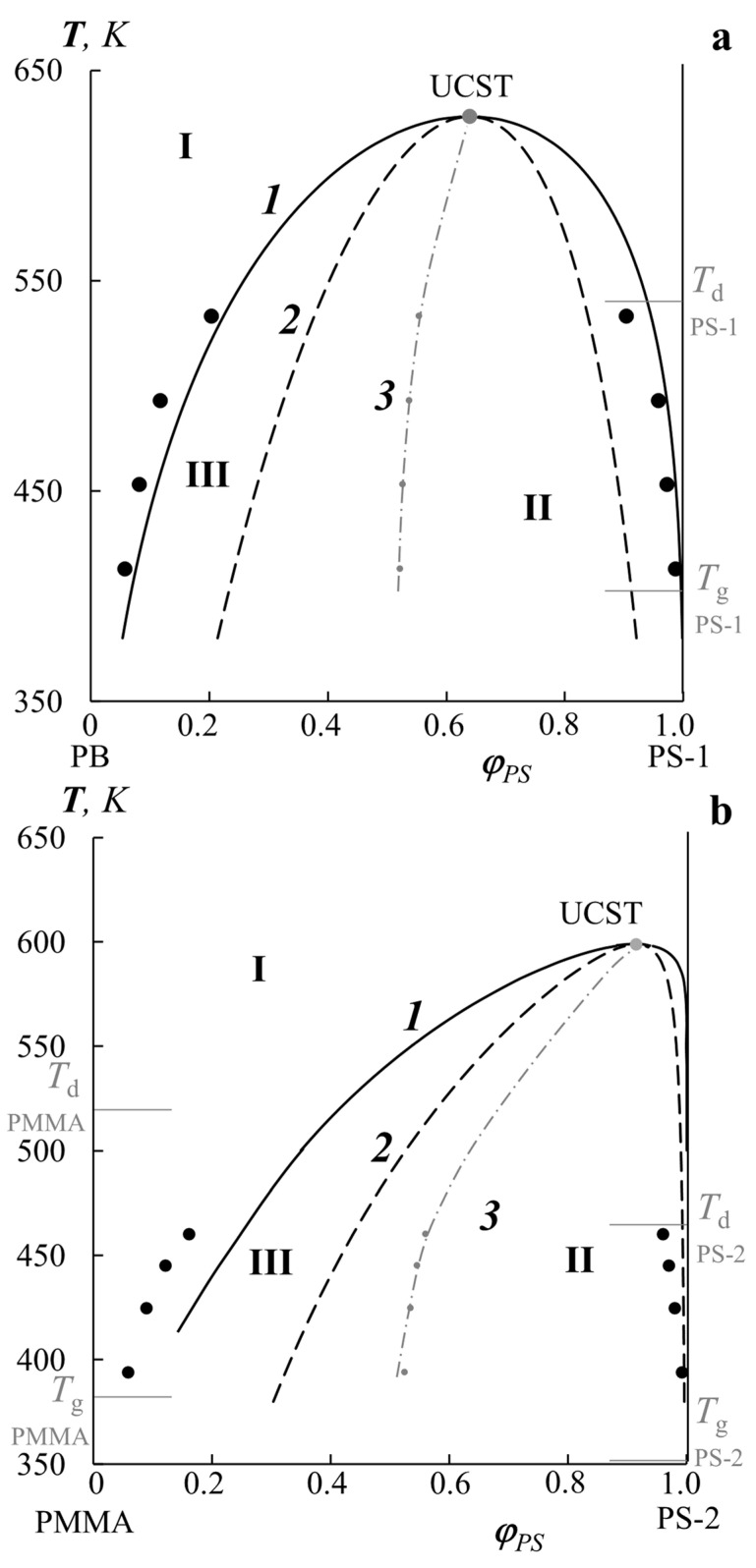
Generalized phase diagrams for PB–PS-1 (**a**) and PMMA–PS-2 (**b**) systems with experimental data on the components’ solubility (black dots), calculated binodal (solid lines 1) and spinodal (dashed lines 2) curves separating the region of homogeneous states (I), the region of labile structures (II), and the region of metastable states (III). Curve 3 corresponds to Alekseev’s diameter. *T*_d_ and *T*_g_ correspond to the degradation and glass transition temperatures of the components, respectively.

## Data Availability

The data presented in this study are available on request from the corresponding author.

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
