# Peer review of "Simulation of Binodal and Spinodal Curves of Phase State Diagrams for Binary Polymer Systems"

_polymers, 2022, doi:10.3390/polym14132524_

Round 1

Reviewer 1 Report

I cannot recommend this paper for publication. I list a number of flaws, problems and doubts:

1) Everything up to line 130 can be find in basic textbooks on Polymers, except the non-properly numbered eqs. (6-9). This part should be greatly summarize to be included in a publication.

2) Figure 3 is confusing. Chemical potencials are in both axes, however the curves do not seem to represent this type of dependence. Since it is an scheme, not a formal graphics, I could understand that the two intersecting curves correspond to values of each potential at different concentrations. In this case, I do not know what the single line above is.

3) Apparently, the numerical method tries to localize critical points in difficult cases where the curves are asymmetrical due to the different molecular weights of the two polymers. This is described in the conclusion, but not in the introduction or the description of the method maintaining the reader confused of the work motivation.

4) The Flory-Huggins parameter should always be refered to a given volume, not specified here. Same problem for the molecular weight proportional r1 and r2 parameters.

5) Apparently, the points in the binodal curves of Figure 4 are directly obtained from the experimental data, but details are not provided about why the data specifically correspond to the thermodynamic phase separation curve.

6) Previous experimental data of the Flory-Huggins parameter obtained by neutron scattering have been reported (see Balsara et al. “Physical properties of polymer handbook”; 2nd ed. New York: AIP: 2007, pages 339-356). For the PMMA-PS system they show a rather flat decreasing dependence with temperature, strongly different that the sharp increased shown in Figure 5. They also are apparently an order of magnitude smaller, though the lack of reference in the present data avoids to be sure about this point.

Author Response

Thank you for your review of our manuscript. We have answered each of your points below.

  1. Everything up to line 130 can be find in basic textbooks on Polymers, except the non-properly numbered eqs. (6-9). This part should be greatly summarize to be included in a publication.

Response: In this article, it was important for us to show in detail the methodology for calculating and constructing a generalized phase diagram, which we consider an important completion in the study of the solubility of the components. The key point is the idea of ​​the possibility of constructing a generalized diagram based on the presence of fragmentary data on the solubility of the components. Such generalized diagrams for the interaction of polymer-polymer pairs are necessary when interpreting the features of the formation of a phase structure in a wide range of work temperatures in technological processes of production and during exploitation, as well as in the synthesis of copolymers. A detailed description of the methodology seemed to us possible only when all the basic equations of the Flory-Huggins theory were presented (and usually there is a question in their absence before the main part). We have made additions to the introduction.

  1. Figure 3 is confusing. Chemical potencials are in both axes, however the curves do not seem to represent this type of dependence. Since it is an scheme, not a formal graphics, I could understand that the two intersecting curves correspond to values of each potential at different concentrations. In this case, I do not know what the single line above is.

Response: Yes, Figure 3 is schematic. It is a graphical illustration of the idea of ​​equality of chemical potentials in determining the composition of coexisting phases. These values ​​of chemical potentials are calculated for a wide range of concentrations (with a small concentration step). The range of intersection is repeatedly refined by reducing the step for the concentrations of the components.

This makes it possible to fairly accurately determine the compositions of coexisting phases even when one of the branches of the binodal curves to the axis (this is often associated with a large difference in the molecular weighs of the components). We have made changes to the text of this part of the article.

  1. Apparently, the numerical method tries to localize critical points in difficult cases where the curves are asymmetrical due to the different molecular weights of the two polymers. This is described in the conclusion, but not in the introduction or the description of the method maintaining the reader confused of the work motivation.

Response: Yes, the technique makes it possible to extrapolate the temperature dependence of the interaction parameter from fragments of binodal curves both to the region of high temperatures and to the region of low ones, and thereby more accurately localize the position of not only critical points, the dome and branches of binodal and spinodal curves, but also the compositions of coexisting phases in difficult cases (for example, when the molecular weights of the components differ greatly). We supplemented this in the introduction and description of the method.

  1. The Flory-Huggins parameter should always be refered to a given volume, not specified here. Same problem for the molecular weight proportional r1 and r2 parameters.

Response: In accordance with D.R. Paul, S. Newman. Polymer Blends. Academic Press, New York-San Francisko-London 1978 (Chapter 2 by Sonja Krause) calculations for the interaction parameter are carried out taking into account the reference volume, usually taken as the molar volume of a repeating unit in the system (in calculations, a reference volume is traditionally used as 100 cm3/mol)

  1. Apparently, the points in the binodal curves of Figure 4 are directly obtained from the experimental data, but details are not provided about why the data specifically correspond to the thermodynamic phase separation curve.

Response: Yes, the solubility data in Figure 4 is derived from experimental data by optical interferometry. This technique makes it possible to obtain information about the entire concentration range at the contact boundary of two components at once. To obtain equilibrium data on solubility, the experiments are carried out in the mode of stepwise heating and cooling with a long exposure at each temperature. The coincidence of these points during heating and cooling allows us to state that the obtained data are in equilibrium. We have made corrections to the experimental part.

  1. Previous experimental data of the Flory-Huggins parameter obtained by neutron scattering have been reported (see Balsara et al. “Physical properties of polymer handbook”; 2nd ed. New York: AIP: 2007, pages 339-356). For the PMMA-PS system they show a rather flat decreasing dependence with temperature, strongly different that the sharp increased shown in Figure 5. They also are apparently an order of magnitude smaller, though the lack of reference in the present data avoids to be sure about this point.

Response: We found and carefully studied the provided reference. Thank you very much for this kind of information. It is undoubtedly very interesting, since it is in the same field of analysis of data on phase diagrams for various molecular weights of components and temperatures. The fundamental difference between our technique can be called an integrated approach to simulation a generalized phase equilibrium diagram (the importance of which was mentioned above) and a different numerical approach to determining the compositions of coexisting phases of binodal curves through the equality of chemical potentials (which is important in complex cases).

Regarding Figure 5 - unfortunately, we made a slight inaccuracy in the designation of the axis (should be 1/T). Accordingly, our data are in good agreement with others. We have corrected the figure and supplemented the text regarding data comparison.

Reviewer 2 Report

see attached file

Author Response

Thank you for your review of our manuscript. We have answered each of your points below.

  1. The paper deals with the determination of the binodal and spinodal curves of binary polymeric solutions. They authors state that the Flory-Huggins theory is able to predict quite accurately equilibrium. The authors explain moreover to have advanced a new („a more efficient and convenient method for numerically determining the compositions of coexisting phases and constructing the position of binodal and spinodal curves“) method of determination of the parameters of the theory. It would be good if they could describe briefly its essence as compared to existing approaches already in the introduction. In detail, it is done in the subsequent section. Obviously, no new assumptions concerning the description of the systems under consideration but the standard relations are used and first briefly repeated. From the outline, I could not conclude why is it preferable to proceed as the authors suggest as compared to previously used approaches. And it remains also unclear for me why the method should be different for systems with upper respectively lower critical solution point. If not, then a similar paper is published by the authors already cited by them as Ref.22.

In addition, there are some bugs in the formulation and the preparation of the figures. Some of them are listed below:

- physic chemistry -> physical chemistry

- Recently, the adapted mean field theory …. -> sentence is not completed

- MethodologY

- Eq.3 can be written in a better form

- ? in Eq.5 (as I suppose) is not defined. Some of the equations are not numbered.

- The quality of the figures should be improved.

- The numbering of the different sections is not performed correctly.

Response: In this article, it was fundamental for us to show in detail the methodology for calculating and constructing a generalized phase diagram, which we consider an important completion in the study of the solubility of the components. Such generalized diagrams for the interaction of polymer-polymer pairs are needed to explain the features of the formation of a complex phase structure obtained during the formation of mixed compositions in a technological process at a wide range of operating temperatures, as well as in the synthesis of copolymers. In our previous articles, we did not specify the features of the numerical calculation for the composition of the coexisting phases of the binodal curve, and we often had questions about this. The classical Flory-Huggins theory cannot predict the existence of LCST in polymer-polymer systems within the framework of its provisions. But we talked about the fact that the mathematical apparatus embedded in the theory allows us to calculate and build generalized diagrams according to our methodology for any systems (both UCST and LCST). We have improved this part of the introduction.

The specified words and sentences have been corrected.

X in equation (5) is used in the average form in accordance with equation (9).

Unfortunately, when the text of the article was transformed into a special format of the publisher, the numbering of equations and chapters suffered. We've fixed it.

Reviewer 3 Report

The paper presents a numerical algorithm for treating binary polymers mixtures based on Flory-Higgins theory.

The manuscript is relatively interesting, although in essence it concerns a numerical method to solve a non linear equation. I wonder about the usefulness of an ad hoc approach for a relatively simple computational task, but I can accept the statement by the authors about its convenience.

Unfortunately the presentation is lengthy and a little confused, with some  missing information (equations numbers !). Editing is required, and  some of the figures are really confusing (e.g. in figure 5 do the author plot 103/T, right?) and missing error bars, where they should be appropriate.

I suggest a possible publication after a general reorganization of the paper.

Author Response

Thank you for your review of our manuscript. We have answered each of your points below.

  1. The paper presents a numerical algorithm for treating binary polymers mixtures based on Flory-Higgins theory.

The manuscript is relatively interesting, although in essence it concerns a numerical method to solve a non linear equation. I wonder about the usefulness of an ad hoc approach for a relatively simple computational task, but I can accept the statement by the authors about its convenience.

Unfortunately the presentation is lengthy and a little confused, with some  missing information (equations numbers !). Editing is required, and  some of the figures are really confusing (e.g. in figure 5 do the author plot 103/T, right?) and missing error bars, where they should be appropriate.

I suggest a possible publication after a general reorganization of the paper

Response: The presented method for calculating binodal and spinodal curves is necessary and convenient for calculating and constructing generalized phase state diagrams based on fragmentary solubility data. Such generalized diagrams for the interaction of polymer-polymer pairs are needed to explain the features of the formation of a complex phase structure obtained during the formation of mixed compositions in a technological process at a wide range of operating temperatures, as well as in the synthesis of copolymers. Moreover, solubility data can be obtained by various methods. Calculations on the example of the presented polymer-polymer pairs showed a good result, which is also presented in the article. We have improved this part of the introduction.

Unfortunately, when the text of the article was transformed into a special format of the publisher, the numbering of equations and chapters suffered. We've fixed it.

On fig. 5 along the axis should, of course, be 1/T (moreover, since these values ​​are small, we took away 10 ^ 3 in the axis label). We've fixed it.

We did not put error bars on the figure, but indicated the measurement error in the experimental part.

Round 2

Reviewer 1 Report

After reading the new version of the manuscript I do not see any significant improvement that can change my previous recommendation. The authors apparently answer all my questions but I cannot see any major change in  the text. I still believe that some basic equations can be suppressed and that the scheme in Figure 3 is not clearly explained. Also, they do not include the reference value for the FH parameter to the text (surprisingly, because they give this "traditional" reference in the answer to my point) and, finally, they claim that their numerical results are similar to the previous ones but they are not (actually, a numerical comparison is not attempted). 

Reviewer 2 Report

I have no further suggestions.

Reviewer 3 Report

//